# Novel Approach for Evaluating Pregnancy-Associated Glycoprotein and Inflammation Markers during the Postpartum Period in Holstein Friesian Cows

**DOI:** 10.3390/ani14101459

**Published:** 2024-05-14

**Authors:** Topas Wicaksono Priyo, Ayane Edo, Yasuho Taura, Osamu Yamato, Tetsushi Ono, Masayasu Taniguchi, Oky Setyo Widodo, Md Shafiqul Islam, Shinichiro Maki, Mitsuhiro Takagi

**Affiliations:** 1Joint Graduate School of Veterinary Science, Yamaguchi University, Yamaguchi 753-8515, Japan; topas.wicaksono@ugm.ac.id (T.W.P.J.); masa0810@yamaguchi-u.ac.jp (M.T.); oky.widodo@fkh.unair.ac.id (O.S.W.); 2Department of Reproduction and Obstetrics, Faculty of Veterinary Medicine, Universitas Gadjah Mada, Yogyakarta 55281, Indonesia; 3Joint Faculty of Veterinary Science, Yamaguchi University, Yamaguchi 753-8515, Japan; dianatomato1212@gmail.com (A.E.); ytaura@yamaguchi.u-ac.jp (Y.T.); yt-ono@yamaguchi-u.ac.jp (T.O.); 4Joint Faculty of Veterinary Science, Kagoshima University, Kagoshima 890-0065, Japan; osam@vet.kagoshima-u.ac.jp (O.Y.); si.mamun@ymail.com (M.S.I.); k6993382@kadai.jp (S.M.); 5Division of Animal Husbandry, Faculty of Veterinary Medicine, Universitas Airlangga, Surabaya 60115, Indonesia

**Keywords:** Holstein Friesian, *FOXP3* gene, pregnancy-associated glycoprotein, serum amyloid A, milk amyloid A

## Abstract

**Simple Summary:**

Pregnancy-associated glycoproteins (PAG) secreted from fetoplacental binucleate cells are used in clinical practice for the early detection of bovine pregnancy. This study examined the postpartum PAG concentration to monitor binucleated cell clearance during postpartum uterine repair. PAG concentration was correlated with inflammatory parameters (serum amyloid A and milk amyloid A), postpartum inflammatory conditions (mastitis, ketosis, and follicular cysts), and *FOXP3* gene-related repeat breeders. Excessive inflammation occurring during the postpartum period may reduce PAG concentration gradually. Clarifying the mechanism of decreased PAG concentration during the postpartum period has a beneficial effect on improving reproductive efficiency, performance, and reproduction aspects.

**Abstract:**

We evaluated the relationship between decreased pregnancy-associated glycoprotein (PAG) levels, inflammatory parameters (serum amyloid A [SAA] and milk amyloid A [MAA]), postpartum inflammatory conditions (mastitis, ketosis, and follicular cysts), and the *FOXP3* gene. Nineteen Holstein-Friesian cows were included in this study. Up to approximately eight weeks after delivery, weekly health examinations were performed for mastitis and ketosis, and reproductive organ ultrasonography was performed. The decreasing PAG rate was negatively correlated with SAA concentration (*r* = −0.493, *p* = 0.032). Cows with mastitis exhibited a slower trend of PAG decrease (*p* = 0.095), and a greater percentage of these cows had MAA concentrations above 12 µg/mL (*p* = 0.074) compared with those without mastitis. A negative correlation, although nonsignificant (*r* = −0.263, *p* = 0.385), was observed between the day-open period and decreased PAG rate. The day-open period was correlated with the presence or absence of follicular cysts (*p* = 0.046). Four cows that developed follicular cysts were homozygous for the G allele of the *FOXP3* gene related to repeat breeders. These results indicate a relationship between a decreased PAG rate and inflammatory status during the postpartum period. Thus, suppressing inflammation during the perinatal period may improve reproductive efficiency in the dairy industry.

## 1. Introduction

Optimal reproductive efficiency significantly affects the dairy industry and is essential for dairy farm management. In cattle, the postpartum period is a crucial window for uterine tissue remodeling, restoration of immunological homeostasis, and resumption of ovarian cyclicity necessary for subsequent fertility [1,2,3]. Field studies have investigated various traits to elucidate the key factors affecting the management of future fertility during the postpartum period in dairy cows, such as dietary management, rearing environments, and disease status of the systemic and genital tracts. Disorders of the metabolic and genital tracts frequently observed during and after calving, such as ketosis, hypocalcemia, endometritis, metritis, mastitis, follicular cysts, dystocia, and retained placenta, which is part of the inflammatory process, delay postpartum uterine recovery, ultimately affecting fertility [4,5,6,7,8,9,10,11,12]. We previously reported the utility of the vaginal discharge score (VDS) in diagnosing clinical endometritis and demonstrated that the VDS grading system improved the reproductive performance of dairy herds [13]. Additionally, the occurrence of VDS Grade 1 (evaluated as mucus-containing flecks of white or off-white purulent material) [13] or calving abnormalities (dystocia, stillbirth, twinning, and retained placenta) may affect reproductive performance [14]. These factors are closely associated with inflammation. Therefore, verifying the relationship between changes in postpartum inflammatory conditions of the genital tract, especially the uterus, and subsequent cattle fertility is essential for improving dairy herd fertility.

Pregnancy-associated glycoproteins (PAG) are aspartic proteinases released into the maternal bloodstream from the binucleate cells (BNC, also known as trophoblast giant cells) of the ruminant placenta during pregnancies [15,16,17,18]. PAG is detected from the 3rd week of early pregnancy until the 10th week postpartum [19,20]. Therefore, PAG is commonly used as a biomarker at practical cattle production sites to detect pregnancy post-artificial insemination and early pregnancy loss [21,22,23,24]. Although the physiological roles of PAG remain unclear, it may help in processing growth factors at the placental-uterine interface, adhesion between the uterus and placenta, or controlling maternal immune modulation [19,25,26]. However, due to the relatively long time required for the clearance of PAG from the maternal blood circulation after calving, the decrease in PAG concentrations to baseline levels exhibits significant variability between females [27,28]. In addition, based on PAG monitoring of postpartum dairy cows with retained placentas, placental retention is associated with higher concentrations of PAG in the maternal plasma at calving and during the postpartum period [29].

Furthermore, the placental tissues of cows with retained placentas have a higher PAG-positive BNC count and PAG mRNA expression than those without retained placentas [17]. Because the detachment of fetal membranes is an inflammatory process, PAG deposited in the maternal septal stroma may contribute to local immunosuppression during pregnancy, and the removal of these inhibitory factors may be a prerequisite for placental rejection [17]. Therefore, the postpartum variability in PAG profiles (i.e., the rate of clearance of BNC from the endometrium) between cows may affect each cow’s uterine recovery status and postpartum reproductive efficacy. Postpartum inflammation occurs in the uterus because of placental expulsion during parturition and postpartum uterine recovery during normal physiological events [30,31]. Peripheral leukocyte populations and functions exhibit alterations during the postpartum period in dairy cattle [2,32], and inflammatory cytokines and complement fragments mediate leukocyte recruitment during inflammation [2,33]. Circulating polymorphonuclear (PMN) cells are activated by proinflammatory cytokines, such as interleukin (IL)-1, IL-6, and tumor necrosis factor-α from monocytes and macrophages, and infiltrate the endometrium led by the chemokine IL-8, where they phagocytize and kill invading bacteria [34]. However, these proinflammatory cytokines are effective stimulators of the production of acute-phase proteins (APPs), such as serum amyloid A (SAA) and milk amyloid A (MAA), which are synthesized in the liver and increase in serum levels in the first few weeks after calving in response to uterine infection [35,36]. Upregulation of APPs and PMN cells is a key feature of uterine inflammation that may be beneficial for successful recovery from inflammation, healthy uterine involution, and a high conception rate [36,37,38]. Therefore, clarifying the relationship between PAG, which is considered a parameter of postpartum uterine repair, and APPs (inflammatory state), which are thought to affect uterine repair, may be important from a pathophysiological aspect of uterine recovery. If such a relationship does exist, controlling postpartum systemic inflammation would lead to favorable uterine recovery. However, to the best of our knowledge, no reports are available on whether the postpartum inflammation status, which may result from systemic negative energy balance (NEB) and bacterial infections of the genital tract or mammary glands that occur during the perinatal period, may affect changes in postpartum PAG concentration.

Regulatory T lymphocyte (T-reg) cells play a role in an immune-tolerant environment during pregnancy [34], T-reg modulates immune regulation during the postpartum period by managing postpartum uterine involution and preparing the genital tracts for the following pregnancy in conjunction with epithelial cell regeneration [39]. *FOXP3* is a master gene that encodes a transcription factor that controls the development and function of T-reg cells; mutations in the gene encoding *FOXP3* have been identified in mice and humans [40]. These mutations may have a detrimental influence on the development of maternal T-reg cells, which may encourage the activation of effector T cells specific to the fetus, leading to infertility [41]. *FOXP3* variants (NC_037357.1: g.87298881A>G, rs135720414) may affect T-reg cell function and are related to negative reproductive performance, such as that in repeat breeders of Japanese Black cattle [41], with a higher G allele frequency of 0.466 in Holstein Friesian dairy herds [42]. No studies have reported a relationship between the postpartum clinical parameters affecting reproductive efficacy and *FOXP3* variants.

This preliminary field trial aimed to evaluate a decreasing rate of PAG concentration in milk (clearance of BNC from the endometrium) that might indicate uterine recovery. A change in PAG concentration will be related to inflammatory parameters, such as SAA and MAA, *FOXP3* variants, subsequent postpartum clinical diseases (mastitis, ketosis, and follicular cysts), and reproductive performance (days open) during the postpartum period. Our findings will provide insights on several factors that inhibit fertility after calving, thereby helping to increase the fertility of dairy cattle.

## 2. Materials and Methods

### 2.1. Animals and Management

This study was conducted on 19 randomly selected primiparous and multiparous Holstein Friesian cows after clinically normal calving (not requiring assistance for calving and no delivery disorders [dystocia, placental retention, prolapsed uterus]; Day 0) from March 2021 to July 2022. Sampling started from weeks 1 to 8 after parturition; sampling for cow numbers 5 and 18 started in the second week, and that for cow number 3 started in the third week. All the cows belonged to one commercial dairy herd in Yamaguchi Prefecture, Japan, where the ambient mean temperature during the sampling period ranged between a minimum of 1.1 °C in January and a maximum of 29.4 °C in July. The herd had approximately 30 lactating cows throughout the year. The herd was non-seasonal and milked twice daily, with an average milk production of 25 to 35 kg/cow per day. The cows received a diet primarily consisting of grass, whole crop silage (WCS) as roughage, and concentrate for dairy cows (dry matter basis: 145–159 g of crude protein/kg and 6.9–7.4 MJ of NEL/kg) during the experimental period. According to the Japanese feeding standards for dairy cattle (Agriculture, Forestry, and Fisheries Research Council Secretariat, 2017), the cows were fed ad libitum to meet their maintenance, growth, and lactation requirements. Feeding was conducted thrice daily with 1 kg Sudan grass, 3 kg Russan, 6 kg total mixed ration (TMR), and 1 kg compound feed in the morning; 1.5 kg oat hay, 4 kg WCS, 6 kg TMR, 1 kg soy pulp, 1 kg flaked corn, and 1 kg mixed feed at noon; and 1 kg Sudan grass, 3 kg Russan, 3 kg WCS, 6 kg TMR, 1 kg soy pulp, and 1 kg mixed feed at night. The parity of the cows ranged from one to six (mean ± SEM; 3.0 ± 0.2; *n* = 19). All the cows were managed in a tie stall with rubber mattresses.

### 2.2. Postpartum Clinical Health Check, Sample Collection, and Farm Assessment

During the study period, the herd was visited once a week every Wednesday, and health checks and clinical examinations were conducted on milk and urine samples of postpartum cows within one week of calving. Briefly, at each cow’s first postpartum visit, after recording the general health examinations (physical assessment, body temperature, heart rate, and respiratory rate), milk from the fourth quarter was collected separately in conical tubes, and urine samples were collected in plastic tubes using a urine catheter. The mastitis status of the milk samples was inspected using the California mastitis test (CMT) (PL tester, Nippon Zenyaku Kogyo Co., Ltd., Fukushima, Japan), according to the manufacturer’s instructions. Briefly, the PL tester was used to determine the pH and number of leukocytes or somatic cells in the milk samples. If no aggregated pieces were observed and the color was golden or yellow in the PL tester, the cow was diagnosed as having no mastitis. To prevent subjectivity, at least two individuals in agreement performed the color interpretation throughout this study. Additionally, in all other cases in which aggregated pieces were confirmed in the samples derived from at least one quarter and the color changed to green in the PL tester, the quarter was diagnosed with mastitis. The diagnosis of mastitis was specific to each quarter. Urine samples were also examined using Multistix urinary strips SG-L (Siemens, Tokyo, Japan) to evaluate ketone status by comparing the colors on the test strip with the color chart of the standards on the package, according to the manufacturer’s instructions. Briefly, the result of the test was obtained within 40 s and scored as follows: negative, 0 mg/dL; negative-positive, 5 mg/dL; positive, 15 mg/dL; positive, 40 mg/dL; positive, 80 mg/dL; and positive, 160 mg/dL. Additionally, the ketosis diagnosis was positive for 15 mg/dL until 160 mg/dL. In the present study, 11 cows were diagnosed with negative results, but all the other cows were diagnosed with ketosis.

For cows diagnosed with ketosis and mastitis at the first medical examination, treatments were administered as prescribed, such as oral administration of propylene glycol preparations or intravenous administration of glucose solutions by drip in cases of ketosis, and intra-mammary administration of sensitive antibiotic ointment in cases of mastitis. Blood and milk sampling was continued every week to confirm improvement in symptoms from the next week onwards; and for the experiment, sampling was conducted until the first postpartum reproductive examination (fresh check). For cows that showed no abnormalities at the first postpartum health check, weekly blood and milk sampling (mixed milk samples from the four quarters for PAG and MAA analyses) were continued until the fresh check. In all cases, blood samples were collected from the coccygeal vein and stored in a heparin tube (Venoject II; Terumo Corp., Tokyo, Japan), and mixed milk from the four quarters was collected for biochemical analysis (Day 1), placed in an ice box containing ice gel, and transported to the Animal Medical Center of Yamaguchi University within 1 h. After centrifugation of all blood and milk samples at 1500× *g* for 15 min, the plasma and whey samples were stored in a freezer at −28 °C until further analysis.

### 2.3. Postpartum Reproductive Management

The voluntary waiting period in the herd was set between 40 and 70 d, depending on the postpartum health condition of each cow, and biweekly follow-ups were conducted for reproductive examinations. The first postpartum reproductive examination (fresh check) of the ovary and uterus included rectal palpation and was confirmed by ultrasound monitoring. During the fresh check, all the follicles with a diameter of >5 mm and corpus luteum (CL) with a diameter of >10 mm were identified via simultaneous ovary monitoring. Our classification of CL in ovarian cows was as follows: (1) presence of CL, (2) absence of CL, and (3) cystic ovaries [43]. We focused on a follicular cyst, an anovulatory follicle with an echogenic area greater than 25 mm that persisted for more than 7 days without a functional CL [44]. The uterus monitoring included paying attention to the fluid in the lumen uterus, in which endometritis, metritis, and pyometra were characterized by a distended uterine lumen with a fluid of specifically echogenic “snowy” patches with the presence of active CL in the ovary [43]. Every week, we observed for purulent or mucopurulent uterine discharge detectable in the vagina after parturition using vaginoscopy [13]. Twelve cows were diagnosed with follicular cysts, and no cows with luteal cysts were identified in the present study.

Additionally, the treatment of reproductive disorders was mainly intramuscular administration of GnRH analog (fertirelin acetate 200 μg), prostaglandin F_2α_ analog (cloprostenol 0.5 mg), and pregnancy diagnosis by one veterinarian using ultrasonography, according to a previous report [43]. Estrous detection was based on hyperemia and swelling of the vulva, mucus discharge, bellowing, restlessness, and/or standing behavior that allowed other cows in the same herd to graze on the playground between the morning and afternoon milking sessions. The cows were inseminated by local artificial insemination (AI) technicians for 8–14 h after estrus detection. Pregnancy was confirmed by transrectal palpation or ultrasonography 40 d after the previous AI. When the pregnancy test confirmed failure to conceive, no hormone treatment was provided for the next cycle. Furthermore, reproductive examinations using ultrasound were continued every two weeks until conception was confirmed. All the examined cows released their placentas normally, and according to clinical symptoms and ultrasound examination, four cows diagnosed with endometritis, metritis, or pyometra were clinically observed after the postpartum check for each cow.

### 2.4. Measurement of Biomarker Concentrations (PAG, SAA, and MAA)

Milk PAG levels were measured using the commercially available Alerts Milk Pregnancy Test (IDEXX Laboratories, Inc., Tokyo, Japan), according to the manufacturer’s instructions. Microtiter plates were coated with anti-PAG antibodies, and the captured PAGs were detected using a PAG-specific antibody in the detector solution. The unbound conjugate was washed, and a tetramethylbenzidine substrate was added as a color indicator to reveal the PAG. After using a stopping solution, the optical density of each well was measured at a wavelength of 450 nm for the sample and controls and at a reference wavelength of 620–650 nm using a microplate photometer (Multiskan FC, Thermo Fisher Scientific, Tokyo, Japan). The results were calculated and expressed as sample negative (S-N) values.

SAA concentration was measured using an automated biochemical analyzer (Pentra C200, HORIBA ABX SAS, Montpellier, France) with an SAA reagent specialized for animal serum or plasma (VET-SAA ‘Eiken’ reagent; Eiken Chemical Co., Ltd., Tokyo, Japan). The SAA concentration was assessed using a calibrator (VET-SAA Calibrator Set; Eiken Chemical Co., Ltd., Tokyo, Japan) and a standard curve of SAA concentration.

MAA concentration was measured using a commercially available ELISA kit (Cow SAA3 ELISA; Veterinary Biomarkers, Inc., West Chester, PA, USA), according to the manufacturer’s instructions. The whey samples were diluted 1:400, and the optical densities were measured using a plate reader at 450 nm. Samples with concentrations over 12.0 µg/mL were further diluted and reanalyzed because the detection limits were 0.75 and 12.0 µg/mL. We skipped the dilution stage because MAA concentrations of 3.87 µg/mL were anticipated according to previous findings [45].

### 2.5. Genotyping of the SNP of the FOXP3 Gene

Blood samples were collected from 19 cows in EDTA tubes (BD Vacutainer; Becton, Dickinson and Company Inc., Franklin Lakes, NJ, USA) and spotted onto Flinders Technology Associates filter papers (FTA card; Whatman International Ltd., Piscataway, NJ, USA). Deoxyribonucleic acid (DNA) was extracted from the disks punched out from blood-impregnated FTA cards following appropriate treatment, as described previously [46]. Genotyping was performed as described previously [42]. Briefly, the primers and TaqMan minor groove binder (MGB) probes used for real-time polymerase chain reaction (RT-PCR) assays were designed based on the sequence of bovine *FOXP3* (NCBI Reference Sequence NC_037357.1).

The sequence of the primers and fluorescent probes used for the RT-PCR was as follows: forward primer, 5′-CCATGTGGCTTCTGAGAAATAGTCA-3′; reverse primer, 5′-TACCTGGAGGGCCAGACT-3′; probe for the A allele, 5′-TCTTCCTGCATTGTCTG-3′; and G allele, 5′-TCTTCCTGCACTGTCTG-3′. These primers and probes, each of which was linked to a fluorescent reporter dye (6-carboxyrhodamine or 6-carboxyfluorescein) at the 5′-end and a non-fluorescent quencher dye at the 3′-end, were synthesized by a commercial company (Applied Biosystems, Foster City, CA, USA). RT-PCR amplifications were performed in 5-µL reactions comprising 2X PCR master mix (TaqMan GTXpress Master Mix; Applied Biosystems) and 80X genotyping assay mix (TaqMan SNP Genotyping Assays; Applied Biosystems) containing the specific primers, TaqMan MGB probes, and template DNA. A negative control containing nuclease-free water instead of the template DNA was included in each run. The cycling conditions were as follows: 20 s at 95 °C, followed by 50 cycles of 3 s at 95 °C and 20 s at 60 °C, with a subsequent holding stage at 25 °C for 30 s. The data obtained were analyzed using StepOne version 2.3 (Applied Biosystems).

### 2.6. Statistical Analysis

Continuous data were checked for normality using histograms. For data that approximated a log-normal distribution, log transformations and other processes were performed to approximate the normal distribution. In cases where it was difficult to approximate the normal distribution via variable transformation, we selected an appropriate non-parametric method.

For each cow, the slope of the degree of decrease in PAG and SAA values with the passage of postpartum days was calculated using a linear mixed model analysis with a random intercept and slope. Because the relationship between days postpartum and PAG and SAA values was not linear but close to log-linear, an estimation model using a logarithmic link function and log-transformed values (natural logarithm) for the dependent variable was constructed, and the model with the best goodness of fit Akaike information criterion (AIC) was adopted. In this model, the log-transformed PAG or SAA values were used as dependent variables, the passage of postpartum days was used as a fixed effect with continuous values, and the subjects were used as a random effect.

Using the model mentioned above, the slope of the log-transformed PAG and SAA values for each cow was estimated as the degree of decrease in PAG and SAA values with the progression of the postpartum period. Furthermore, each cow’s SAA value (log-transformed) for Day 1 was estimated using the above mixture model and was used as an indicator. Scatter plots of days postpartum and PAG values were plotted to calculate the rate of decrease in PAG values (Figure 1a,b). Some cows showed a significant increase (*p* < 0.05) in PAG values; however, a general monotonically decreasing trend was observed, especially in the log-transformed PAG values, which showed a linearly decreasing trend. Therefore, a mixed-effects model was used to estimate the daily PAG reduction for each cow.

Correlation analysis of SAA and PAG (log-transformed measurement and slope [rate of decrease], respectively) within a sample was performed using a linear mixed model that included individual cows with fixed effects (PAG as the dependent variable and SAA as the independent variable). Correlations were calculated by analyzing SAA and PAG using standardized variables with log-transformed values.

Pearson’s product-rate correlation analysis was performed to evaluate the correlation among continuous variables, such as PAG slope, log-transformed SAA value, and postpartum days open period of the cows. Welch’s *t*-test was also used for correlations, such as the PAG slope with mastitis, follicular cysts, and ketosis. Fisher’s exact test was used to evaluate the correlations between nominal measures. A general linear and additive model was used to determine the association between *FOXP3* variants and mastitis, follicular cysts, and the number of open days. For all analyses, statistical significance was set at *p* < 0.05. Statistical analyses were performed using the SPSS software (version 24.0; IBM Corp., Armonk, NY, USA).

## 3. Results

### 3.1. First Evaluation: Relationship between Postpartum Day and Biomarkers (PAG and SAA)

The characteristics of the analyzed cows are shown in Appendix A. For all cows in the analysis (*p* < 0.001), the rates of decrease in PAG concentration, log-transformed PAG, and log-transformed SAA were −0.085/day, −0.062/day, and −0.036/day, respectively (Table 1). By comparing the PAG concentration before (with the link function set to logarithmic) and after (with the link function set to linear) log transformation, we found that estimating the reduction using log-transformed PAG values produced a model with a lower AIC value.

Finally, based on the statistical analysis findings, we developed the following formulas to calculate the values of PAG, log-transformed PAG, and log-transformed SAA.
PAG = exp (1.749 − 0.085 × days postpartum)  Log. PAG = 5.061 − 0.062 × days postpartum  Log. SAA = 2.550 − 0.036 × days postpartum

Description of exp: the relative likelihood of model *i* (relative likelihood comparison between different values of a parameter in a single model).

### 3.2. Second Evaluation: Correlation between the Postpartum Rate of Decrease in PAG and SAA

We observed a moderately negative correlation between the rate of decrease in SAA and PAG values (*r* = −0.493, *p* = 0.032; Figure 2). In contrast, the within-subject correlation between log-transformed SAA and PAG on each measurement day was significantly positive (*r* = 0.346, *p* = 0.003; Figure 3). A non-significant but positive correlation was found between the day-one estimate of SAA and PAG slope (*r* = 0.413, *p* = 0.079). No significant correlation was observed between the mean SAA value and the rate of PAG decrease during the follow-up period (*r* = 0.092, *p* = 0.718). No correlation was observed between the PAG slope and postpartum days open (*r* = −0.263, *p* = 0.385; Figure 4).

### 3.3. Third Evaluation: Association between Each Parameter with and without Mastitis, Follicular Cyst, and Ketosis

Appendix A shows the associations between each parameter in cows with and without mastitis during the experimental period. Statistical tendencies were observed between cows with or without mastitis on the PAG slope (*p* = 0.095) and cows with MAA concentrations greater than 12 μg/mL (*p* = 0.074). The results of the evaluation of the association between the presence or absence of follicular cysts, and each parameter are presented in Table 2. A significant difference (*p* = 0.046) in the postpartum open periods was observed between cows with and without cysts. Appendix A shows the results of parameter comparisons between cows that developed ketosis or urinary ketones (*n =* 8) and those that did not (*n =* 11). No significant differences (*p* > 0.05) were detected in any of the parameters.

### 3.4. Fourth Evaluation: Association between FOXP3 Variants (T-Reg Genotypes) with Cyst, Mastitis, and Postpartum Days Open

An evaluation of the association between *FOXP3* variants (T-reg genotypes) and cysts, mastitis, and postpartum days open showed that the G/G genotype was associated with a significantly higher incidence of cysts (*p* = 0.017; with an additive model; Table 3). In contrast, although no significant difference (*p* > 0.05) was observed, an increase in the incidence of mastitis and number of days of open periods was observed in the order of the A/A, A/G, and G/G genotypes.

## 4. Discussion

Our field study indicated a correlation between decreased postpartum PAG concentration and inflammatory markers, such as SAA, in dairy cattle. This is the first report indicating the relationship between postpartum PAG and SAA concentrations. Furthermore, some aspects of postpartum uterine recovery may be associated with the development of follicular cysts and *FOXP3* variants.

PAG concentration is widely used clinically as a biomarker for early pregnancy diagnosis in bovine blood and milk samples [47]. However, the characteristics of PAG secreted from the placental BNC have been used to validate the usefulness of PAG concentration as a screening method for monitoring postpartum reproductive failure, such as placental retention occurring immediately after parturition or early detection of metritis and endometritis that develop during the perinatal period, which greatly affects the fertility of cattle [17,29]. However, to the best of our knowledge, no previous study has used the postpartum rate of decrease in PAG expression (PAG slope) as a diagnostic parameter for postpartum uterine recovery. The present study focused on the relationship between postpartum PAG expression in milk and systemic inflammatory biomarkers associated with uterine involution, as well as the incidence of puerperal diseases, such as mastitis and metabolic ketosis, in the postpartum period and their relationship with reproductive efficacies, such as the occurrence of follicular cysts and postpartum day open periods. The main results of our field trial were as follows: (1) a significant positive correlation (*p* < 0.001) between PAG and SAA concentrations on each measurement day; (2) although not significant (*p* = 0.139), a positive correlation between SAA concentration and PAG slope on postpartum Day 1 (*r* = 0.343); (3) a moderate negative correlation between the rate of decrease in SAA value and that in PAG value (*r* = −0.493, *p* = 0.032); (4) a significant difference (*p* = 0.046) in postpartum days open period among cows with cysts and those without cysts; with having the *FOXP3* variant being associated with a significantly higher incidence of cysts (*p* = 0.017), and finally, (5) no correlation between the rate of decrease in PAG and postpartum days open periods (*r* = −0.263, *p* = 0.385). Despite the limited number of animals examined, we demonstrated a significant positive correlation between postpartum PAG and SAA concentrations, the possibility of postpartum PAG variation, *FOXP3* allele variation, and the occurrence of follicular cysts.

Because detachment of fetal membranes can be regarded as an inflammatory process, PAG may contribute to local immunosuppression during pregnancy, and removing these inhibitory factors may be a prerequisite for placental rejection [17]. After parturition, the plasma PAG concentration decreased significantly in maternal blood or milk until 7–14 d postpartum; subsequently, the PAG concentration gradually declined and reached its lowest level at 70–90 d postpartum [19,28,47,48]. Active inflammation occurs during not only the expulsion of the placenta during parturition but also postpartum uterine involution, which is a normal physiological process [30,49]. In contrast, some reports have indicated that in cows with a retained placenta, the PAG concentration in the maternal serum during the postpartum period is higher than that in control cattle, suggesting that it is causally involved in the development of the retained placenta by suppressing the normal immunological processes of the detachment of fetal membranes [17,29].

First, a significant positive correlation between the PAG and SAA was observed on each measurement day. This suggested that the higher the postpartum day 1 SAA concentration (higher inflammation), the gentler the slope of the PAG, whereas the lower the postpartum day 1 SAA concentration (lower inflammation), the steeper the slope of the PAG. This indicates the following relationships: (1) if the initial SAA value is high, the PAG slope is gentle; (2) if the initial SAA value is high, the SAA slope is steep (the higher the initial value, the faster the rate of decrease tends to occur); conversely, when the initial SAA value is low, the SAA slope is gentle during the postpartum period, which is attributed to small changes in SAA concentrations with minimal inflammation. From the above, if the initial value of SAA (estimated day 1 SAA) is considered as an axis, “the slope of PAG is gentle” and “the slope of SAA is steep” may be established, and the slopes may exhibit a negative correlation (Figure 2). In simple terms, a lower SAA concentration (low inflammatory status) indicates a steep decrease in PAG concentration; the SAA concentration may influence PAG concentration during the postpartum period.

As the occurrence of typical early postpartum disease seems to be associated with impairment of the immune system in dairy cows during early lactation, we evaluated its relationship with the occurrence of clinical diseases, such as mastitis, follicular cysts, ketosis, PAG slope, postpartum day open period, and inflammation markers (SAA and MAA). As shown in Appendix A, statistical tendencies were observed between cows with or without mastitis on the PAG slope (*p* = 0.095) and cows with MAA concentrations greater than 12 μg/mL (*p* = 0.074). This observation suggests that cows without mastitis tend to have faster PAG decreases, indicating a potential influence of MAA, rather than SAA, on the slope of PAG decline in cows affected by mastitis. Table 2 shows a significant difference (*p* = 0.046) in the postpartum days of the open periods between the cows with and without follicular cysts. Additionally, no correlation was observed between follicular cysts and the SAA or MAA parameters, indicating that the degree of inflammation resulting from mastitis or systemic inflammation and/or the rate of postpartum PAG decrease may not have a direct effect. Finally, no significant differences (*p* > 0.05) were observed in any of the parameters examined between cows with or without ketosis (Appendix A). This study conducted a general health examination within one week of delivery, and urine and milk samples were used to diagnose mastitis and ketosis under home (i.e., farm) conditions. Because mastitis and ketosis treatments were started immediately for the cattle diagnosed as positive, the two groups may not have exhibited significant differences in the various parameters due to the effects of the treatment. In contrast, follicular cysts were diagnosed 1–2 months after calving, when reproductive health checkups were initiated; therefore, a significant difference (*p* < 0.05) was observed in the number of days open between cows with and without follicular cysts. Thus, to clarify the relationship between uterine recovery status based on postpartum PAG concentration and frequent postpartum mastitis and ketosis, it is necessary to conduct intensive controlled studies at the research laboratory level rather than the farm level.

In recent years, uncovering the role of maternal immune regulatory cells and their regulatory genes that may also contribute to infertility has gained increasing attention in veterinary research. In particular, the development, differentiation, and immunosuppressive functions of T-regs have been the focus of numerous studies since the identification of *FOXP3* [41,42]. We detected cows with *FOXP3* variants G/G (*n* = 4) and A/G (*n* = 13), but not A/A (*n* = 2; Table 3). Recently, we reported the frequency of the G allele of the *FOXP3* variant in four cattle breeds (Japanese Black, Holstein, Korean Hanwoo, and Indonesian Madura) and found that the frequency in Holstein cattle was 0.466 (*n* = 73) [42]. Interestingly, significant differences (*p* = 0.017) in follicular cysts (A/A: 0% and G/G: 100%) were observed among the three groups; however, the limited number of cows in the present study prevents this result from being fully generalizable. However, this study presents basic information, and the researcher may conduct a future study about the relationship between *FOXP3* and follicular cysts. Additionally, although no significant differences (*p* > 0.05) were observed, an increase in the incidence of mastitis and the number of days of the open period were observed in the order of A/A, A/G, and G/G genotypes. A previous report suggested high SAA values in cows with ovarian cysts [50]. In the present study, although no significant differences (*p* > 0.05) were observed in the estimated Day 1 SAA and SAA slopes between cows with and without follicular cysts, the G allele of the *FOXP3* variant may affect the function of T-reg cells, thereby affecting the different inflammation statuses between cows with and without follicular cysts and possibly with mastitis and days of open periods. Further investigations of fertility and disease morbidity in cattle with the G allele of *FOXP3* may be essential to further improving herd productivity. Furthermore, we recently reported a significant negative correlation between fluctuations in APPs (SAA and the albumin/globulin ratio) and anti-Müllerian hormone (AMH), a marker of the ovarian reserve of small antral follicles and a possible fertility marker during the peripartum period in dairy cows [3]. These data suggest that elucidating the mechanism of the inflammatory effects that strongly influence perinatal AMH changes and the suppression of inflammation during the perinatal period may help improve reproductive efficiency in the dairy industry.

## 5. Conclusions

In conclusion, our field study indicated a correlation between decreased PAG concentration and inflammation status during the postpartum period in dairy cattle. We assumed that a rapidly decreasing concentration of PAG indicated low inflammation, whereas a gradually decreasing concentration of PAG indicated a heightened inflammatory status during the postpartum period. Furthermore, some aspects of decreased postpartum PAG concentration may suggest a relationship between the development of follicular cysts and the *FOXP3* variant. Further studies are necessary to investigate the relationship between postpartum uterine recovery and PAG/SAA/*FOXP3*, which may affect the reproductive efficiency of dairy cattle.

## Figures and Tables

**Figure 1 animals-14-01459-f001:**
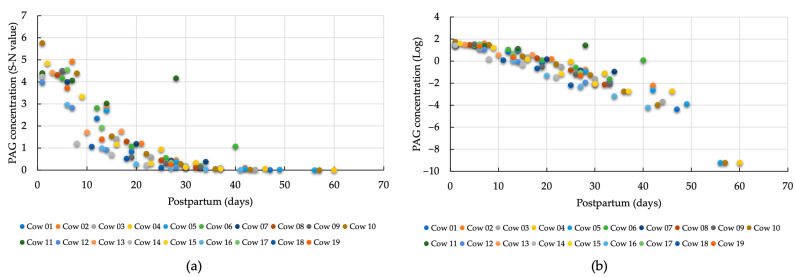
Comparison of PAG concentrations: (**a**) before (with the link function set to logarithmic); (**b**) after (with the link function set to linear) log-transformation. S-N value, sample-negative value; Log, log-transformation; PAG, pregnancy-associated glycoprotein.

**Figure 2 animals-14-01459-f002:**
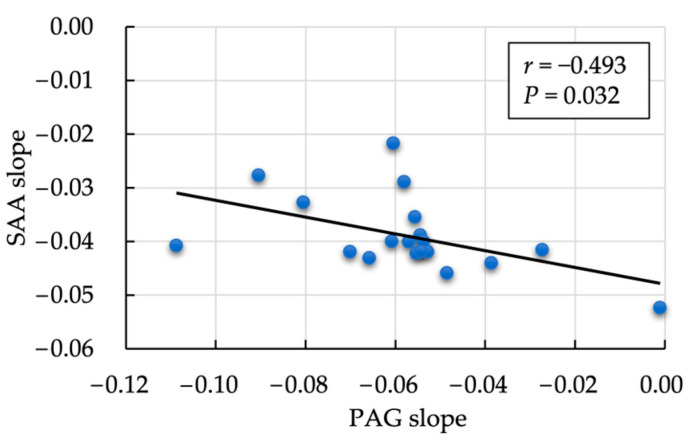
Correlation between the postpartum rate of decrease in pregnancy-associated protein (PAG) and serum amyloid A (SAA) values. Correlation analysis of PAG and SAA within each sample was evaluated using a linear mixed model that included individual cows as the fixed effects. The correlation was calculated by analyzing SAA and PAG using standardized variables with log-transformed values.

**Figure 3 animals-14-01459-f003:**
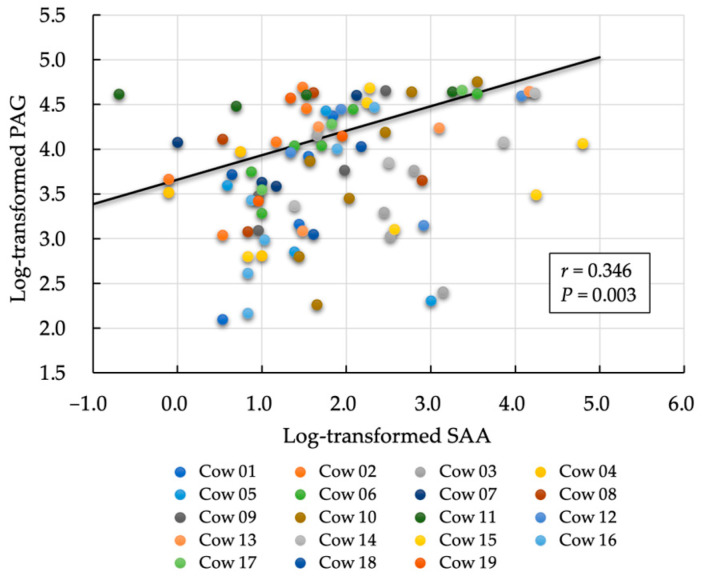
Within-subject correlation between pregnancy-associated protein (PAG) and serum amyloid A (SAA) on each measurement day using a mixed model analysis.

**Figure 4 animals-14-01459-f004:**
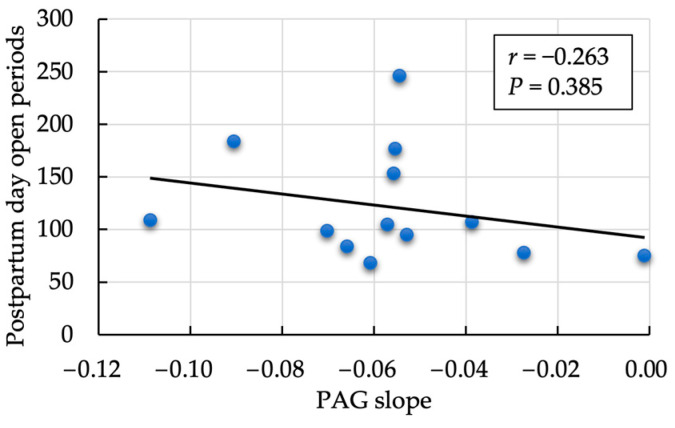
Correlation of pregnancy-associated protein (PAG) slope and postpartum days open period.

**Table 1 animals-14-01459-t001:** Relationship between the postpartum day and biomarkers.

		Estimate	95% CI		*p*-Value	AIC
PAG	Segment	1.749	1.648	1.850		238.329
	Postpartum day	−0.085	−0.097	−0.073	<0.001 *	
Log. PAG	Segment	5.061	4.882	5.239		115.193
	Postpartum day	−0.062	−0.069	−0.056	<0.001 *	
Log. SAA	Segment	2.550	2.118	2.983		246.075
	Postpartum day	−0.036	−0.050	−0.022	<0.001 *	

CI, confidence interval; *, significant difference; AIC, Akaike information criterion; log, log-transformed.

**Table 2 animals-14-01459-t002:** Association between follicular cysts, PAG slope, postpartum days open period, and inflammation markers.

	Follicular Cyst		*p*-Value
	without	with	with vs. without
No. of cows	7	12	
Days open period (mean ± SD)	94.5 ± 14.3	140.4 ± 60.6	0.046 *
PAG slope	−0.061 ± 0.033	−0.052 ± 0.021	0.524
Estimate day 1 SAA	2.81 ± 1.20	2.36 ± 9.22	0.367
SAA slope	−0.040 ± 0.007	−0.040 ± 0.005	0.860
No. of cows with MAA more than 12 µg/mL (%)	4 (57.1)	8 (66.7)	0.999

SD, standard deviation; PAG, pregnancy-associated glycoprotein; SAA, serum amyloid A; MAA, milk amyloid A; *, significant difference, *p* < 0.05.

**Table 3 animals-14-01459-t003:** Relationship between the *FOXP3* variants, follicular cyst, mastitis, and postpartum days open.

	*FOXP3* Variant			
	A/A	A/G	G/G	*p*-Value
No. of cows	2	13	4	
Follicular cyst (%)	0 (0)	8 (61.5)	4 (100)	0.017 *
Mastitis (%)	0 (0)	8 (61.5)	3 (75.0)	0.135
Days open period (mean ± SD)	102.0 ± 9.9	119.9 ± 50.9	154.8 ± 74.5	0.239

SD, standard deviation; *, significant difference, *p* < 0.05.

## Data Availability

The original contributions presented in the study are included in the article/Appendix A, and further inquiries can be directed to the corresponding author.

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
