# Peer review of "Novel Approach for Evaluating Pregnancy-Associated Glycoprotein and Inflammation Markers during the Postpartum Period in Holstein Friesian Cows"

_animals, 2024, doi:10.3390/ani14101459_

Round 1

Reviewer 1 Report

Comments and Suggestions for Authors

Priyo TW Jr et al. reported a novel approach for evaluating pregnancy-associated glycoprotein and inflammation markers during the postpartum period in holstein friesian cows.

The presented manuscript is interesting and covers issues that have not been studied until now. I am particularly impressed by the detailed description of Materials and Methods.Nevertheless, I noticed a few points that require clarification or addition.

1. What device was used for colorimetric measurements of PAG concentration in milk?

2. What was the positive control in the PAG milk test?

3. Why was the PAG concentration not determined in peripheral blood, which was collected? Especially since PAG concentrations in cow's milk are much lower (20 to 30 times) than in serum [Gajewski et al. 2008; Friedrich and Holtz 2008]. In goats and sheep, differences in PAG concentrations in plasma and milk are much smaller.

4. How much gDNA template was used for the FOXD3 genotyping reaction?

5. PAG concentration testing was performed only after delivery and only in milk. It is a pity that the concentration of PAG in the blood during pregnancy was not tested, perhaps it would be a basis for explaining the single individuals with high PAG values after delivery demonstrated by the authors. I understand that this cannot be done and added to the results presented in this manuscript, but perhaps in future studies. The individual difference could also result from other reasons. It is known that there is a correlation between the concentration of PAG proteins in plasma and the sex of the fetus [Zoli et al. 1992; Patel et al. 1998] or single and multiple pregnancies [Barbato et al. 2008].

6. The results of FOXD3 genotyping and the prediction of the occurrence of ovarian cysts, mastitis or an increase in days of open periods are interesting. This constitutes the basis for further research and, perhaps in the future, a test for selecting cattle and improving reproductive results.

Author Response

Reviewer 1

Comments and Suggestions for Authors

  1. What device was used for colorimetric measurements of PAG concentration in milk?

Accordingly, we have added information on the colorimetric device to the Materials and Methods section. Lines 233-234.

  1. What was the positive control in the PAG milk test?

In this study, using the positive control sample provided with the commercial kit, we calculated the S-N value (S-N value ≥ 0.250; pregnancy) for a positive result in full, according to the kit instructions.

  1. Why was the PAG concentration not determined in peripheral blood, which was collected? Especially since PAG concentrations in cow's milk are much lower (20 to 30 times) than in serum [Gajewski et al. 2008; Friedrich and Holtz 2008]. In goats and sheep, differences in PAG concentrations in plasma and milk are much smaller.

Thank you very much for this insightful comment. We agree with your opinion. As one of the objectives of the present field study was to verify the association between changes in PAG concentration and development of mastitis diagnosed using milk samples, we developed a plan using a milk PAG measurement system in the present study.

  1. How much gDNA template was used for the FOXP3 genotyping reaction?

We extracted gDNA from a 1.2-mm diameter disc punched out of blood-impregnated FTA cards using a special solution (DNA Extract All Lysis Reagents Kit; Applied Biosystems) included in the commercial kit (TaqMan GTXpress Master Mix; Applied Biosystems), as described previously. The concentration of gDNA in the DNA solution was not measured because the extraction method was simple. However, the amount of gDNA extracted using this simple method was sufficient for genotyping. No errors or failures were observed in any of the genotyping assays. This procedure has been described in detail in reference no. 46, and we have mentioned it in the original manuscript.

  1. PAG concentration testing was performed only after delivery and only in milk. It is a pity that the concentration of PAG in the blood during pregnancy was not tested, perhaps it would be a basis for explaining the single individuals with high PAG values after delivery demonstrated by the authors. I understand that this cannot be done and added to the results presented in this manuscript, but perhaps in future studies. The individual difference could also result from other reasons. It is known that there is a correlation between the concentration of PAG proteins in plasma and the sex of the fetus [Zoli et al. 1992; Patel et al. 1998] or single and multiple pregnancies [Barbato et al. 2008].

Thank you for your insightful suggestions for future research. We hope to develop future research with your comments in mind and clarify the details of our analysis of the factors affecting postpartum uterine recovery in dairy cows.

  1. The results of FOXP3 genotyping and the prediction of the occurrence of ovarian cysts, mastitis or an increase in days of open periods are interesting. This constitutes the basis for further research and, perhaps in the future, a test for selecting cattle and improving reproductive results.

Thank you for the suggestion. Accordingly, we have added an explanation for the correlation about FOXP3 correlation with follicular cysts for further research. Lines 486-489.

Reviewer 2 Report

Comments and Suggestions for Authors

General comments

In this manuscript, the authors reported their study on relationships between pregnant-associated glycoprotein(PAG) and postpartum inflammation. The methods were reasonably good and the results are interesting. However, the information generated was limited and the conclusion was not fully supported by the results. The data about the effect of FOXP3 variants on follicular cyst and mastitis are interesting, but the sample size was not enough to generate reliable conclusions.

Special comments

1.      Lines 34-37. This sentence needs to be revised as the results were not conclusive (P > 0.05).

2.      Lines 39-41. The results were only based on the association between PAG and SAA. The correlation between PAG and other inflammatory features, such as mastitis, was not significant. Other inflammatory parameters, such as inflammatory cytokines and prostaglandins could be investigated to strengthen the conclusion.

3.      Lines 112-114. Rephrase.

4.      Line 138. Were the cows sampled on same day in each week?

5.      Lines 166 and 169. Change “sample” to “samples”.

6.      Lines 255-256. It looks that these forward and reverse primers do not amplify bovine FOXP3. Please check the sequences.

7.      Statistical data analysis. Did you check the distribution of the data. If the data do not follow a normal distribution, a non-parametric correlate analysis should be tried.

8.      Lines 286-287. How could a mixed effect model be used on only one cow (each cow) as cow ID should be a random effect in this case?

9.      Lines 292-296. In the linear mixed effect model, cows should be random effect rather than fixed effect.

10. Line 324. Which log-transform was used, log10 or natural log? If log10 was used, this should be 10 powered rather than "exp" which is used for natural log transform.

11. Figure 3. Which time point (day) were the values of SAA and PAG derived from?

12. Line 453. As the correlation was not significant (P>0.05), significant tendencies could not be claimed.

13. Lines 458-462. Rephrase.

14. Lines 509. As the correlation between and mastitis was not significant, this was not conclusive.

Comments on the Quality of English Language

A moderate editing of English writing is required.

Author Response

Reviewer 2

Comments and Suggestions for Authors

Special comments

  1. Lines 34-37. This sentence needs to be revised as the results were not conclusive (P > 0.05).

Accordingly, we have revised the sentence as follows: “Cows with mastitis exhibited a slower trend of PAG decrease (P = 0.095) and a greater percentage of these cows had MAA concentrations above 12 µg/ml (P = 0.074) compared with those without mastitis.” Lines 33-35.

  1. Lines 39-41. The results were only based on the association between PAG and SAA. The correlation between PAG and other inflammatory features, such as mastitis, was not significant. Other inflammatory parameters, such as inflammatory cytokines and prostaglandins could be investigated to strengthen the conclusion.

Thank you for your insightful comments. As mentioned in the Introduction, this is a challenging field study conducted based on our new conception. Based on the novel significant results and trends obtained from this study, it will be necessary to increase the number of cases in the future, taking into account the comments from the reviewer. We hope to develop future research with your comments in mind, such as on inflammatory cytokines and prostaglandins, and to clarify the details of our analysis of the factors affecting postpartum uterine recovery to increase the reproductive efficacy of dairy herds. Accordingly, we revised the sentence: "These results indicate a relationship between a decreased PAG rate and inflammatory status during the postpartum period. Thus, suppressing inflammation during the perinatal period may improve reproductive efficiency in the dairy industry.” Lines 39-41.  

  1. Lines 112-114. Rephrase.

Accordingly, we have rephrased the sentence: "T-reg modulates immune regulation during the postpartum period by managing postpartum uterine involution and preparing the genital tracts for the following pregnancy in conjunction with epithelial cell regeneration.” Lines 110-112.

  1. Line 138. Were the cows sampled on same day in each week?

We apologize for the lack of explanation. Following your comment, we have added an explanation for the samples collected daily. Lines 154.

  1. Lines 166 and 169. Change “sample” to “samples.”

We have revised this accordingly. Lines 163 and 167.

  1. Lines 255-256. It looks that these forward and reverse primers do not amplify bovine FOXP3. Please check the sequences.

Thank you for the suggestion. We rechecked these issues based on the NCBI database. These primers amplify the location from 87298844 to 87298928 (85 base pairs) on NC_037357.1, which is "Bos taurus isolate L1 Dominette 01449 registration number 42190680 breed Hereford chromosome X, ARS-UCD2.0, whole genome shotgun sequence." The location of the SNP (g.87298881A>G) inside this region was 2,175 bp upstream of the start codon of bovine FOXP3 exon 1. This information is described in reference no. 42.

  1. Statistical data analysis. Did you check the distribution of the data. If the data do not follow a normal distribution, a non-parametric correlate analysis should be tried.

Thank you for your comments. As stated in the Statistical Analysis in the Methods section, in this study, log transformations and link functions are made to log links as necessary after checking the distribution. We adopted a parametric approach, with a treatment in which all data distributions were approximated to a normal distribution. In accordance with your suggestion, we have added the following sentence at the beginning of the document, which is a bit more explicit: “Continuous data were checked for normality using histograms. For data that approximated a log-normal distribution, log transformations and other processes were performed to approximate the normal distribution. In cases where it was difficult to approximate the normal distribution via variable transformation, we selected an appropriate non-parametric method.” Lines 274-278.

  1. Lines 286-287. How could a mixed effect model be used on only one cow (each cow) as cow ID should be a random effect in this case?

In this study, the analytical model included a random intercept and slope in the mixed model. Because a random slope was included, it was possible to estimate the slope for each cow using linear estimation after the mixed model.

  1. Lines 292-296. In the linear mixed effect model, cows should be random effect rather than fixed effect.

As stated in the original manuscript, cattle are included in the random effect. We added sentence as follows; “and the subjects were used as a random effect”. Line 287.

  1. Line 324. Which log-transform was used, log10 or natural log? If log10 was used, this should be 10 powered rather than "exp" which is used for natural log transform.

We used the natural log. We have followed your suggestion and added this information to the Methods section.

  1. Figure 3. Which time point (day) were the values of SAA and PAG derived from?

In this study, the time point (day) starts on Day 1.

  1. Line 453. As the correlation was not significant (P>0.05), significanttendencies could not be claimed.

We apologize for the lack of awareness of this issue. We have revised the sentence as follows; “statistical tendencies”. Line 455.

  1. Lines 458-462. Rephrase.

According to this remark, we have rephrased the sentence as follows; “Table 2 shows a significant difference (P = 0.046) in the postpartum days open periods of cows with and without follicular cysts. Additionally, no correlation was observed between follicular cysts and the SAA or MAA parameters, indicating that the degree of inflammation resulting from mastitis or systemic inflammation and/or the rate of postpartum PAG decrease may not have a direct effect.” Lines 459-464.

  1. Lines 509. As the correlation between and mastitis was not significant, this was not conclusive.

Thank you for your insightful comments. Accordingly, we have revised the conclusion to clarify the decreased PAG concentration relationship with follicular cysts and FOXP3 as follows; “Furthermore, some aspects of decreased postpartum PAG concentration may suggests a relationship between the development of follicular cysts, and FOXP3 variants”. Lines 511-513.

Comments on the Quality of English Language

A moderate editing of English writing is required.

The revised version of the manuscript underwent English language editing by Editage.

Reviewer 3 Report

Comments and Suggestions for Authors

The study offers new opportunities to understand the role of PAG in the apparition/resolution of inflammatory process during the postpartum period. Nevetherless attending the huge number of factors influencing the days open, it’s difficult to understand the possible relation between PAG and this parameter.  Such evaluation of relation don’t have a true interest.

 I have some difficulties to understand the selected parameters (ketosis, follicular cysts, mastitis) to evaluate the effect of PAG.

Unfortunately, no comparison group (for exemple cows with a placental retention) has been selected.

Moreover, very few parameters have been described to evaluate the quality of uterine recovery (clinical signs, neutrophils population…).

The discussion on the relations between PAG and SAA or MAA concentrations and « inflammatory » pathologies are rather difficult to understand.

30 as inflammatory conditions you are speaking on ketosis and mastitis. In line 23 you are speaking on follicular cysts and mastitis : why this difference ? Could you explain why you consider ovarian cysts as an inflammatory condition ? Can you explain why you consider ketosis as an inflammatory process ?

53 may be  you can add also hypocalcemia in relation with neutrophils activity

66 most in not necessary

111 can you mention that Regulatory T celles are lymphocytes ?

115 can you explain the possible relation between the Fo P3 gene variant and the pathogeny of cysts. Moreover, I do’nt undertand the relation between PAG concentrations, inflammatory markers and the FoP3 gene

135 primiparous or multiparous cows ? Uterine recovery can be different with the number of lactation

136 have you selected only the cows with no placental retention ?

156 how have you excluded the presence of acute or puerperal metritis during the period of evaluation ?

164 what’s a PL Tester ?

167 the interpretation of the color can be subjective

168 how many quarters need to be positive to consider the cow as having a mastitis ?

175 how these different cutt-off have been determined ?

177 Can you explain this sentence : In the present study, the cows were diagnosed as having no 177 ketosis in the negative case, but all other cases were diagnosed as having ketosis. attending

203 what about the luteal cyst ?

269 as a clinician it’s rather difficult to understand the statistical used methodology

306 the results of MAA concentrations are not presented except to describe the relation with pathologies. Why ?

315 can you mention number of cows with metritis (acute/puerperal) and endometritis

315 is it possible to have the first AI pregnancy rate ?

351 have you analyzed the possible cumulative effect between pathologies and PAG or SAA concentrations ?

366 table 2 you mention 12 cows with cysts but 11 cows with cyst in table S1

389 and what about the relation with MAA ?

425 What are the differences between your PAG concentrations and the results of other studies.

433 these informations don’t have been mentionned in the results

447 Finally what kind of relation between PAG and SAA can be proposed ? PAG concentrations influence the SAA and MAA concentrations or SAA and MAA concentrations influence the PAG concentrations ?

Author Response

Reviewer 3

Comments and Suggestions for Authors

  1. The study offers new opportunities to understand the role of PAG in the apparition/resolution of inflammatory process during the postpartum period. Nevetherless attending the huge number of factors influencing the days open, it’s difficult to understand the possible relation between PAG and this parameter.  Such evaluation of relation don’t have a true interest.

Thank you for your thoughtful comment. We were the first to report the relationship between the onset of placental retention and PAG concentration (arrest of binucleate cells in the uterus) in cows, and we first hypothesized that PAG concentration might be useful in monitoring the status of uterine recovery. We hypothesized that a rapid decrease in PAG concentration would affect postpartum uterine recovery, which would significantly impact postpartum reproductive efficiency. We examined the relationship between inflammation-associated diseases such as mastitis and ketosis. Based on the pathophysiological results obtained from our field survey, we would like to proceed with research on new prevention and treatment methods that promote the early recovery of the uterus of dairy cows after calving, thereby increasing the reproductive efficacy of dairy herds. We hope that you understand this.

  1. I have some difficulties to understand the selected parameters (ketosis, follicular cysts, mastitis) to evaluate the effect of PAG.

Thank you for this insightful comment, and we apologize for the difficulty in understanding. As mentioned above, we aimed to evaluate (1) whether the decreasing rate of PAG concentration might be useful as an indicator of uterine recovery, and (2) whether the decreasing rate of PAG concentration correlates with postpartum inflammatory diseases such as mastitis, ketosis, and follicular cysts, which are frequently observed during the postpartum period. We have added and revised the Introduction to explain the purpose in greater detail. Lines 53-55; 123-129.

  1. Unfortunately, no comparison group (for exemple cows with a placental retention) has been selected.
  2. Moreover, very few parameters have been described to evaluate the quality of uterine recovery (clinical signs, neutrophils population…).

Thank you for your comments. As mentioned in the Materials and Methods section, samples from cows with normal calving were obtained during the test period; however, no cows developed retained placenta, so samples from cows with retained placenta could not be included.

Also, regarding the parameters to evaluate the quality of uterine recovery, in the present study, we could monitor the fluid in the lumen uterus, in which endometritis and pyometra are characterized by a distended uterine lumen with a fluid of specifically echogenic “snowy” patches with the presence of active CL in the ovary, and every week, we observe purulent or mucopurulent uterine discharge detectable in the vagina after parturition using vaginoscopy. Finally, because we believe the parameter of “postpartum days open” might be one of the parameters of reproductive efficacies of the dairy herds under the same management condition, the number of postpartum days open was used as a parameter for uterine recovery. As the reviewer pointed out, examining the relevance of other parameters in the future will be necessary. Thank you for your comments.

  1. The discussion on the relations between PAG and SAA or MAA concentrations and « inflammatory » pathologies are rather difficult to understand.
  2. Line 30 as inflammatory conditions you are speaking on ketosis and mastitis. In line 23 you are speaking on follicular cysts and mastitis : why this difference? Could you explain why you consider ovarian cysts as an inflammatory condition? Can you explain why you consider ketosis as an inflammatory process?

Thank you for your thoughtful comments. We apologizes for the confusion about the sentences of inflammatory conditions due to the lack of information we referred to. We referred to previous reports [Abuajamieh M.; Kvidera, S.K.; Fernandez, M.V. ; Nayeri, A.; Upah, N.C.; Nolan, E.A.; Lei, S.M.; DeFrain, J.M.; Green, G.B.; Schoenberg, K.M.; Trout, W.E.; Baumgard, L.H. Inflammatory biomarkers are associated with ketosis in periparturient Holstein cows. Res. Vet. Sci. 2016, 109, 81–85] in the case of ketosis and [Stassi, A.F.; Díaz, P.U.; Gasser, F.B.; Velázquez, M.M.L.; Gareis, N.C.; Salvetti, N.R.; Ortega, H.H.; Baravalle, M.E. A review on inflammation and angiogenesis as key mechanisms involved in the pathogenesis of bovine cystic ovarian disease. Theriogenology 2022, 186, 70–85] in cases of follicular cysts has been newly added to the reference list. We revised the parts as follows: “postpartum inflammatory conditions (mastitis, ketosis, and follicular cysts)”. Lines 22, 28-29, 53-56, 123-129.

  1. Line 53 may be you can add also hypocalcemia in relation with neutrophils activity

Accordingly, we have added “hypocalcemia” to the sentence referred to previous reports [Zhang, B.; Ma, X.; Loor, J.J.; Jiang, Q.; Guo, H.; Zhang, W.; Li, M.; Lv, X.; Yin, Y.; Wen, J.; et al. Role of ORAI calcium release-activated calcium modulator 1 (ORAI1) on neutrophil extracellular trap formation in dairy cows with subclinical hypocalcemia. J. Dairy Sci. 2022, 105, 3394–3404]. Lines 53-54.

  1. Line 66 most is not necessary

Accordingly, we deleted it.

  1. Line 111 can you mention that Regulatory T celles are lymphocytes?

Accordingly, we have revised it to “Regulatory T lymphocytes (Treg)”. Line 109.

  1. Line 115 can you explain the possible relation between the FOXP3 gene variant and the pathogeny of cysts. Moreover, I don’t understand the relationship between PAG concentrations, inflammatory markers and the FOXP3 gene

Thank you for your comments. The FOXP3 gene regulates the growth and activity of T-regs, which are closely associated with the immune system during inflammatory responses. Follicular cysts are a reproductive disorder with an inflammatory basis; thus, we aimed to clarify the relationship between FOXP3 and follicular cysts. In the present study, we examined the relationship between PAG concentrations and inflammatory markers and between PAG concentrations and FOXP3, which are closely associated with the inflammatory response. We have revised and added information to clarify the relationship between PAG, an inflammatory marker, and the FOXP3 gene. Lines 459-464, 486-489, 511-513.

  1. Line 135 primiparous or multiparous cows? Uterine recovery can be different with the number of lactation

We used a mixture of primiparous and multiparous cows in the present study. A previous report showed no significant difference between primiparous and multiparous dairy cows during the uterine involution period (Dai et al., 2023). We have added the following references: Dai, T.; Ma, Z.; Guo, X.; Wei, S.; Ding, B.; Ma, Y.; Dan, X. Study on the pattern of postpartum uterine involution in dairy cows. Animals 2023, 3, 3693. We have added sentences and explanations to the text. Lines 132-133.

  1. Line 136 have you selected only the cows with no placental retention?

As mentioned above, in this study, we used cows with normal calving and without disordered delivery (dystocia, retention of the placenta, and prolapsed uterus). We have added and revised the explanation of normal calving without delivery disorders. Lines 134.

  1. Line 156 how have you excluded the presence of acute or puerperal metritis during the period of evaluation?

Accordingly, in this study, we used all samples of postpartum uterine infections with clinical signs of vaginal discharge, such as endometritis, metritis, and pyometra. We monitored reproductive postpartum using rectal palpation, ultrasound, and vaginoscopy as described in the Materials and Methods section. We have added a sentence describing metritis to the Materials and Methods section. Line 204.

  1. Line 164 what’s a PL Tester?

We apologize for any confusion regarding the PL tester. The PL tester was a product of the California Mastitis Test in Japan. Accordingly, we revised this part of the manuscript. Line 161.

  1. Line 167 the interpretation of the color can be subjective

Thank you for this insightful remark. We agree with your concern. On the other hand, the method is a very simple and useful diagnostic tool in clinical practice. To

  1. Line 168 how many quarters need to be positive to consider the cow as having a mastitis?

In the present study, at least one teat or quarter that changed color and aggregates was diagnosed as mastitis. We apologize for the confusion regarding the word teat and have changed teat to quarter. Lines 158, 167-169.

  1. Line 175 how these different cut-offs have been determined?

Thank you for your comments. Accordingly, we have added an explanation for eliminating the diagnosis of ketosis. Lines 174-175.

  1. Line 177 Can you explain this sentence: In the present study, the cows were diagnosed as having no ketosis in the negative case, but all other cases were diagnosed as having ketosis.

We apologize for the confusion regarding the sentences in the Ketosis section. We have revised it as follows; “Additionally, the diagnosis was positive for positive 15 mg/dL until 160 mg/dL. In the present study, 11 cows were diagnosed with negative cases, but all other cows were diagnosed with ketosis.” Lines 174-177.

  1. Line 203 what about the luteal cyst?

Thank you for your insightful comments. In the present study, no cows were diagnosed as luteal cysts; no classification as luteal cysts was mentioned in the Materials and Methods section. Lines 208-209.

  1. Line 269 as a clinician it’s rather difficult to understand the statistical used methodology

Thank you for pointing this out. We believe that our statistical methods are the best way to evaluate all results obtained in this study. Thank you for your comments.  

  1. Line 306 the results of MAA concentrations are not presented except to describe the relation with pathologies. Why?

In this study, we presented and evaluated MAA concentrations in relation to the results of postpartum inflammatory conditions such as mastitis and ketosis.  

  1. Line 315 can you mention number of cows with metritis (acute/puerperal) and endometritis

Thank you for your comments. We have added the number and percentage of cows with endometritis, metritis, or pyometra. Line 326.

  1. Line 315 is it possible to have the first AI pregnancy rate?

Following your comment, we have added the pregnancy rate with a successful first artificial insemination (AI). Line 326.

  1. Line 351 have you analyzed the possible cumulative effect between pathologies and PAG or SAA concentration?

Thank you for your insightful comments. Because this study is the first and basic investigation of the relationship between PAG and inflammation markers and postpartum inflammatory conditions (mastitis, ketosis, follicular cysts), we did not analyze the possible cumulative effect between pathologies and PAG or SAA concentrations in the present study. The points you raised are important for further investigation.  

  1. Line 366 table 2 you mention 12 cows with cysts but 11 cows with cyst in table S1

We apologize for the lack of awareness of this issue. Accordingly, we checked the number of cows and revised them accordingly. Line 326.

  1. Line 389 and what about the relation with MAA?

Accordingly, in the first paragraph of the Discussion section, we explain only the parameters that significantly affect the results to make it easier for the reader to understand our paper. We do not mention the MAA in the first paragraph; however, the MAA results are not significantly different.

  1. Line 425 What are the differences between your PAG concentrations and the results of other studies.

We investigated the relationship between postpartum PAG concentration and inflammatory markers, especially SAA and MAA, as well as their relationship with inflammatory conditions during the postpartum period. In this study, we emphasize and focus on the novelty of our trial; to our knowledge, there are no other similar reports.

  1. Line 433 these informations don’t have been mentionned in the results

Thank you for your comments. Following your comment, we moved these sections to the Materials and Methods section of the revised manuscript. Lines 221-224.

  1. Line 447 Finally what kind of relationship between the PAG and SAA is proposed. PAG concentrations influenced SAA and MAA concentrations or the influence of SAA and MAA concentrations on PAG concentrations.

Thank you for the thoughtful comment. Accordingly, we apologize for the confusion regarding the relationship between PAG and SAA. We have added the following explanation to clarify this part: “In simple terms, a lower SAA concentration (low inflammatory status) indicates a steep decrease in PAG concentration; the SAA concentration may influence PAG concentration during the postpartum period”. Lines 448-450.

Round 2

Reviewer 2 Report

Comments and Suggestions for Authors

In this revised manuscript, the authors incorporated or addressed most of my comments. Its quality is now improved. I have one minor comment.

 1. Line 460. Change “of cows” to “between the cows”.

Comments on the Quality of English Language

English writing is acceptable. 

Reviewer 3 Report

Comments and Suggestions for Authors

Thank you to have improve your paper according to the remarks.